# Effect of Mg Addition and PMMA Coating on the Biodegradation Behaviour of Extruded Zn Material

**DOI:** 10.3390/ma16020707

**Published:** 2023-01-11

**Authors:** Alia A. Diaa, Nahed El-Mahallawy, Madiha Shoeib, Nicolas Lallemand, Flavien Mouillard, Patrick Masson, Adele Carradò

**Affiliations:** 1Design and Production Engineering Department, Faculty of Engineering, Ain Shams University, Cairo 11517, Egypt; 2Department of Design and Production Engineering, Faculty of Engineering and Materials Science, German University in Cairo, Cairo 11835, Egypt; 3Central Metallurgical Research and Development Institute, El Tebbin, Cairo 11722, Egypt; 4Institut de Physique et Chimie des Matériaux de Strasbourg, IPCMS, UMR 7504 CNRS, Université de Strasbourg, 67000 Strasbourg, France

**Keywords:** alloys, biodegradation, coat, extrusion, zinc, grafting-from method, PMMA

## Abstract

Although zinc (Zn) is one of the elements with the greatest potential for biodegradable uses, pure Zn does not have the ideal mechanical or degrading properties for orthopaedic applications. The current research aims at studying the microstructure and corrosion behaviour of pure Zn (used as a reference material) and Zn alloyed with 1.89 wt.% magnesium (Mg), both in their extruded states as well as after being coated with polymethyl methacrylate (PMMA). The grafting-from approach was used to create a PMMA covering. The “grafting-from” method entails three steps: the alkali activation of the alloys, their functionalization with an initiator of polymerization through a phosphonate-attaching group, and the surface-initiated atom transfer radical polymerisation (SI-ATRP) to grow PMMA chains. Electrochemical and immersion corrosion tests were carried out in a simulated body fluid (SBF), and both confirmed the enhanced corrosion behaviour obtained after coating. The electrochemical test revealed a decrease in the degradation rate of the alloy from 0.37 ± 0.14 mm/y to 0.22 ± 0.01 mm/y. The immersion test showed the ability of complete protection for 240 h. After 720 h of immersion, the coated alloy displays minute crevice corrosion with very trivial pitting compared to the severe localized (galvanic and pitting) corrosion type that was detected in the bare alloy.

## 1. Introduction

Attempts have been made to develop biodegradable metals that are expected to degrade steadily in vivo with a proper host reaction. The corrosion by-products of the biodegradable metals should be released so that cells may digest, metabolize, or excrete them. Meanwhile, the biodegradable metal implant should completely disintegrate after completing its role in supporting the injured tissues [1].

Increased research interest in zinc (Zn) over other biodegradable metals is to achieve a more controlled rate of biodegradation compared to that of magnesium (Mg) and iron (Fe) without hydrogen bubble formation or the release of harmful residues or ions [1,2].

Due to its low strength and high brittleness, cast pure Zn is seldom employed as a biodegradable metal. However, to overcome the lack of mechanical integrity and to regulate the degradation rate, many processes are being developed, such as alloying, thermomechanical deformation, and surface modifications [2]. The most common Zn alloy systems consist of Zn-Mg-, Zn-Li-, Zn-Mn-, Zn-Ag-, and Zn-Cu-based alloys [3]. Due to the microgalvanic corrosion between the intermetallic phase and the matrix, the strengthening effect of the multiphase Zn alloy usually comes at the expense of the degradation behaviour. Thermomechanical processing focuses mainly on grain refining by dynamic recrystallization and fragmentation of intermetallic phases [4,5,6].

Mostaed et al. [7] studied a variety of Zn-Mg alloys with Mg weight percentages ranging from 0.15 to 3. It was observed that the attained ultimate tensile strength (UTS) ranges between 250 MPa and 399 MPa, the elongation (El) ranges between 22% and 1%, and the corrosion rate ranges between 130 μm/y and 0.17 μm/y. For the alloys with UTS greater than 300 MPa, the UTS of the alloys Zn-3Mg and Zn-1Mg was approved for bone and stent implants; nevertheless, the suitable range of corrosion rates for orthopaedic applications must vary from 0.169 μm/y to 0.125 μm/y (extracted from the graph) [8].

It is essential to control the degradation of the metallic biomaterial in order to avoid problems with local alkalization. It can cause haemolysis of the red blood cells and high metallic ion release, which results in low cell viability and the rapid loss of mechanical strength before the biomaterial can fulfil its supporting function [9,10]. Phosphate conversion coating, biomimetic deposition of Ca- and P-rich layers, organic and polymer coating, anodic oxidation, micro-arc oxidation, atomic layer deposition of different metallic oxides, magnetron sputtering of carbon films, and sandblasting were all discussed for a more controlled biodegradation behaviour of Zn-based materials [11].

Shomali AA [12] investigated the biodegradation and biocompatibility of pure Zn wire (0.25 mm) coated with a polymeric coat of poly (L-lactic acid) (PLLA). 3-(trimethoxysilyl)propyl methacrylate (MPS) silane was used to improve adhesion by forming a chemical bond between PLLA and zinc. The electrochemical study of 1 mm thickness of PLLA coating on pure Zn revealed a 16-fold increase in impedance when compared to bare metal, indicating a lower corrosion rate for the coated Zn. Furthermore, the long-term in vivo biodegradation in the abdominal aorta of a rat implies that the coating delays the biodegradation processes by approximately 6 months compared to the uncoated pure Zn. Shenghui Su [13] proposed calcium phosphate conversion coatings followed by doping a polylactic acid (PLA) with lithium octacalcium phosphate particles in order to enhance the biodegradability and biocompatibility of pure Zn. After coating, the corrosion current density (I_corr_) decreased from 8.26 × 10^–5^ A/cm^2^ of pure zinc to 6.56 × 10^–6^ A/cm^2^, and the impedance rose from ~336 Ω cm^2^ to 1200 Ω cm^2^, indicating that the polymeric coat represents an efficient strategy for reducing the corrosion rate.

Polymethylmethacrylate (PMMA) coatings applied to decrease the biodegradability of metallic materials were biologically studied and found to have a good cell attachment and viability [14,15]. PMMA coatings proved to be efficient for a significant decrease in the high affinity of degradation of Mg-based substrates, and they were applied through many procedures. Majumder [16] used an alkali treatment followed by dipping in silane (SiH4) solution and then PMMA solution to introduce silane-PMMA as a coating on AE42 Mg alloy. Jin [15] also applied the PMMA coat using spin coating of a ZK60 alloy in PMMA solution in anisole, while Gonsalves [17] used a simple immersion tetrahydrofuran (THF) solution in applying the PMMA-co-PMAA (poly (methacrylic acid)) coating with different copolymer percentages on pure Mg. Nevertheless, the spin-coating technique [15] does not allow a stable covalent bond between the polymer and the alloy. To develop a stable grafting of PMMA chains on a titanium (Ti) substrate, Reggente et al. [14] employed a surface-initiated atom-transfer radical polymerization (SI-ATRP) process. The PMMA coating was applied through the “grafting from” method, which involves three steps: (i) activation by alkali treatment, followed by (ii) functionalization with the initiator, namely bromoisobutyrate-undecyl-1-phosphonic acid (C_15_H_30_O_5_PBr), and finally (iii) growth of the polymeric chain from the initiator via ATRP [18,19]. Rather than other methods, such as the sol–gel techniques, the main advantage of the “grafting from” method is that it introduces strong chemical bonds between the polymeric chain and the functionalized alloy. In other words, the polymeric coat here was chemically bonded to the Ti surface and not just bonded by physical interactions, as is achieved with other techniques. This “grafting from” method resulted in thicker, denser PMMA layers grafted to the Ti surface.

For the current investigation, a hot extruded Zn-Mg alloying system was selected; this choice was motivated by our prior research on the microstructural characteristics, mechanical properties, and biodegradability [6]. Among the different polymeric coating processes, the PMMA coating developed by Reggente et al. [14] on Ti substrate was applied to the Zn-Mg alloy.

The present manuscript deals with the preparation and characterization of Zn-Mg alloy coated with PMMA, and the evaluation of its strength and durability during the corrosion process in the biological medium of the covalent bond between Zn-1.89Mg and PMMA. The PMMA/Zn-1.89Mg is designed by using an ATRP method according to our previous works and in view of biomedical applications [14].

In this work, two objectives are targeted: the formation of a continuous PMMA coating by chemical bonding and the good behaviour of the PMMA/Zn-1.89Mg towards behaviour corrosion. Furthermore, the current study suggests a novel technique of PMMA coating to modify the corrosion mechanism and consequently enhance its biodegradation resistance. However, the coating is permanent because PMMA is not biodegradable. The development of this grafting method on Zn alloys allows us to prove the design. In the future, we plan to apply this method to other polymers, such as bioresorbables.

## 2. Materials and Methods

### 2.1. Materials Processing and Characterization

Pure Zn and an alloy with a nominal composition of Zn-1.89Mg were prepared from pure Zn (>99.9 wt.%) and Mg ingots (>99.9 wt.%). The chemical analysis of the alloys (weight percent) was performed using inductively coupled plasma-atomic emission spectrometry (ICP-AES Varian 715-Es, Varian spectroscopy, Palo Alto, CA, USA). Casting was carried out through melting in an electric resistance furnace at 500 °C (Zn melts at 419 °C) in a pottery crucible put in an electrical furnace, and after ensuring a complete fusion of Zn, Mg was added while stirring for 5–7 min to ensure a homogenous composition of the melt. Afterwards, the melt was cast in a permanent steel cylindrical mould (about ø 55 × 72 mm) preheated to 200 °C. The ingots were left to cool down at room temperature. The cylindrical ingots of Zn-1.89Mg were heat treated at 350 °C for 8 h. The heat-treated ingots were machined to a diameter of 50 mm for removing the oxide skin. The extrusion equipment used was a 100-ton-capacity hydraulic press. The extrusion system is heated to 320 ± 5 °C by an electric coil attached to it and isolated from the press. Hot extrusion was carried out at the same temperature after stabilising the system for about 15 min. The extrusion speed was set at 4 mm/s. The extruded rod diameter was about 11 mm, representing an extrusion ratio of 16:1.

Microstructure figures were taken by an Olympus BX51 P microscope, Japan. Phase identification was available through the Riga D/max 2500PC X-ray diffractometer (XRD) using a Cu Kα radiation (λ = 0.1789 nm, 40 kV, 40 mA) and a scanning rate 4° of per min.

### 2.2. Grafted PMMA on Zn-1.89Mg

The specimens were cut from the extruded bars into discs of 10 mm diameter and 1 mm in thickness. They were mounted on copper cylinders before being ground with a P4000 grit SiC abrasive paper, and polished with diamond paste (1PS4, ESCIL). After that, they were ultrasonically rinsed with cyclohexane, acetone, ethanol, and deionized water for 10 min each. Samples were suspended via Teflon wire to avoid contact between them and left to dry under a fume hood. Surfaces were activated through alkali treatment by immersion in a 5 M sodium hydroxide (NaOH) solution stirred and heated at 80 °C in a Teflon beaker. After 1 h, the discs were washed with deionized water. The activated samples were immediately immersed in a flask containing an aqueous solution of 2 mM bromoisobutyrate-undecyl-1-phosphonic acid (C_15_H_30_O_5_PBr, previously synthesised in the lab) and stirred and heated at 95 °C in the dark. After 24 h, the discs were ultrasonically washed with dichloromethane and deionized water for 10 min each. The PMMA was grown according to ATRP reaction via a grafting-from technique described in previous work [14] as follows: the functionalized discs were put in a reactor equipped with a magnetic stirrer; this reactor was degassed through three argon-vacuum cycles, followed by the addition of 39 mg of copper bromide (Cu(I)Br) (Sigma Aldrich, St. Louis, MO, USA), 18 mL of anisole (Acros Organics, Waltham, MA, USA), and 48 μL of pentamethylene diethyl-triamine (PMDETA) (TCI). The reactive medium was stirred to yield a homogeneous solution. Then, 18 mg of malononitrile (TCI) was included to enhance the polymerization rate. Finally, 106 mmol of MMA (Acros Organics)—purified from its stabiliser by column chromatography on basic aluminium oxide (Sigma Aldrich)—was added to the reactor, and the reaction was carried out under argon at 35 °C. The quantity of monomer was optimized in order to obtain the highest amount of monomer in a 22 mL and 5.3 M concentrated solution. After 24 h, discs were rinsed with methanol and ultrasonically cleaned with deionized water in order to remove unreacted monomer and catalytic residues.

### 2.3. PMMA Coating Characterization

Fourier transform infrared (FTIR) spectroscopy: FTIR spectroscopy in total reflectance geometry was used in order to conduct the chemical characterization and identify the functional groups present on the Zn-1.89Mg samples’ surface after functionalization and coating. This characterisation was carried out with the Perkin Elmer Spectrum Two, Waltham, MA, USA, working with wavelengths between 4000 and 400 cm^−1^, 32 scans, and a resolution of 4 cm^−1^.

Scanning electron microscopy: the coated samples’ surface composition and morphology were studied through a scanning electron microscope (SEM), JSM-6700F, JEOL, Tokyo, Japan, set to use the information given by backscattered electrons and equipped with an energy-dispersive X-ray spectrometer (EDX).

Cross-sectional study: For the analysis of the Zn-1.89Mg/PMMA interface, the samples’ edges were bevelled by grinding, then coated with Cu using plasma deposition to finally pass through an ion beam cross polisher (Hitachi IM4000+, Japan). The ion beam used was an argon-ionised plasma with a current of 130 µA. This preparation allowed the cross-section to be observed using the SEM Zeiss, Geminis SEM 500, Oberkochen, Germany.

### 2.4. Electrochemical Measurements

Electrochemical tests (potentiodynamic polarisation and electrochemical impedance spectroscopy (EIS)) were performed on both uncoated Zn and coated Zn-1.89Mg specimens. The uncoated surfaces were prepared for corrosion testing through grinding and polishing to be mirror-like. Simulated body fluid (SBF) was chosen as a corrosion medium. The SBF was prepared with the following composition: 8.035 g/L NaCl, 0.355 g/L NaHCO_3_, 0.225 g/L KCl, 0.231 g/L K2HPO_4_·3H_2_O, 0.311 g/L MgCl_2_·6H_2_O, 39 mL 1.0 mol/L HCl, 0.292 g/L CaCl_2_, 0.072 g/L Na_2_SO_4_, 6.118 g/L Tris. These components were added in the order listed, and 5 mL of 1.0 mol/L HCl was gradually added to achieve a pH of 7.4 [20]. Potentiodynamic and electrochemical impedance measurements using a potentiostat/galvanostat were used for the electrochemical measurements in accordance with ASTM G5-87. The software Nova 1.10 was used for data analysis, fit, and simulation. Potentiodynamic polarisation and EIS studies were configured in a cell using three electrodes assembly: A Ag/ AgCl reference electrode, a Pt counter electrode, and coupons-like specimens of subjected surface of 0.196 cm^2^ as the working electrode in a SBF corrosion medium at 37.0 ± 0.5 °C. The system was allowed to perform an open circuit potential (OCP) with a steady-state behaviour working range in the region of 400 mV below and 600 mV above OCP, a scan rate (1 mV/s), and a range of frequencies (kHz). EIS was performed at frequencies ranging from 100 mHz to 100 kHz. The amplitude of the sinusoidal voltage signal was 10 mV.

### 2.5. Static Immersion Test

The immersion test was carried out in SBF solution (as specified in Section 2.4. Electrochemical Measurements) according to ASTM-G31-72. The solution volume to the sample surface area was between 20 to 40 mL/cm^2^ [21]. Six immersion time periods were selected for experimentation: 6, 24, 120, 240, 480, and 720 h. Each sample was weighed before and after each immersion time and the corrosion products were removed using a gentle mechanical brushing. Afterwards, the corrosion rate was calculated using the equation [21]:(1)$$\mathrm{Corrosion}\;\mathrm{rate}\;(\mathrm{mm}/y)=\frac{8.76\times 10^{7}\times \;\mathrm{weight}\;\mathrm{loss}\;\left(g\right)}{\mathrm{surface}\;\left({\mathrm{cm}}^{2}\right)\times \;\mathrm{exposure}\;\mathrm{period}\;\left(h\right)\times \;\mathrm{density}(\mathrm{gm}/{\mathrm{cm}}^{3})}$$

### 2.6. Characterization of Surfaces after Corrosion

FEI Inspect S50, Czech Republic, scanning electron microscope (SEM) equipped with EDX and EBSD analysis with Esprit 1.8.5 software was used. A carbon sticky pad was employed to stick samples to the SEM table. The stereoscopic 2.5D photos of the corroded surfaces after removing the corrosion products and their topography were obtained from the Carl Zeiss microscope, Germany, using ZEN 2012 software (blue edition) and ImageJ software (V.1.48, NIH).

### 2.7. Statistical Analysis

All experimental numerical values were obtained by calculating the means of at least three readings and are stated with a ± standard deviation

## 3. Results

### 3.1. Microstructure of Pure Zn and Zn-1.89Mg

From the microstructure of as-extruded pure Zn and Zn-1.89Mg, the shape and size of grains reveal the effect of alloying elements on the tendency of grains to be recrystallized by hot extrusion. In pure Zn, a mixed grain size was observed, including large grains with noticeable size differences and other small grains (Figure 1a). The large grains were non-uniform in shape and had sizes close to the initial grains before extrusion, ranging from 31 µm to 335 µm. The small grains were equiaxed with an average size of 11 ± 5 µm. The proportion of the small grains attained after extrusion was 23%, indicating the percentage of the partial thermomechanical recrystallization that occurred. The addition of 1.89Mg was capable of decreasing the size of the non-recrystallized grains (which range from 3 µm to 16 µm) and achieving a higher percent of recrystallized grains of 55% with an average size of 1.2 ± 0.6 µm (Figure 1b). The large difference in grain size between pure Zn and Zn-1.89Mg alloy translates into a different scale for the two microstructures in Figure 1. This confirms Jarzębska’s findings [22] that the addition of Mg induces the dynamic recrystallization of pure Zn. A detailed illustration of the grain size distribution related to the corresponding area fraction is given in the Appendix A.

The Zn-1.89Mg alloy comprises two phases, a soft α-Zn phase (Figure 2, point 1) and a solid solution phase surrounded by an α-Zn+Mg_2_Zn_11_ eutectic structure (Figure 2, point 2). The two-phase eutectic structure represents about 25.6 ± 5.3%. The eutectic structure is mainly composed of Mg_2_Zn_11_ lamellate structure, representing 81.3 ± 5.8% of the surface area, fully surrounding fine areas of α-Zn solid solution, with an average size of 1.3 ± 0.7 μm. EDX data are given in the Appendix A.

XRD examination verified the alloy’s multiphase structure (Figure 3). The composition of the lamellar eutectic structure is arguable. While the peaks of the present study’s XRD confirmed the presence of the Mg_2_Zn_11_ phase and no peaks of the MgZn_2_ phase were detected, Ye [23] claims that the eutectic structure of as-cast Zn-1.5Mg contains nanoparticles of the MgZn_2_ intermetallic phase in addition to the Mg_2_Zn_11_ intermetallic phase. The absence of the nanostructured metastable MgZn_2_ main phase may be attributed to the alloy’s heat treatment and thermomechanical working.

### 3.2. Surface of PMMA-Coated Zn-1.89Mg

The “grafting from” PMMA coating aims at creating a strong adherence between the alloys and the PMMA through a covalent bond. It starts with the alkali activation, which results in altering the metallic surface’s composition and morphology. New hydroxyl groups are formed on the surface of the metal with hierarchical porosity [14]. The hydroxyl group endorses the grafting process, while the porosity permits polymerization to happen in a columnar manner. The presence of the hydroxyl groups is confirmed by the ATR-FTIR spectra (Figure 4), peak at 3438 cm^−1^. After that comes the grafting of the surface through immersion in the bromoisobutyrate-undecyl-1-phosphonic acid initiator (C_15_H_30_O_5_PBr) at the conditions stated in the experimental section. Grafting produces a PMMA chain with one end of the phosphonic acid group linked to the metal oxide layer and the other end of the PMMA chain containing the ATRP initiator’s bromo ester group. The eleven-hydrocarbon ([C11]) spacer between the phosphonate and reactive bromo ester groups allows one group’s reactivity to be independent of the other. As evidence for the success of grafting, the peaks corresponding to the initiator bonds in their pure form should be detected. The FTIR detects the peaks of asymmetric and symmetric vibrations of the methylene group (CH_2_) at 3000 cm^−1^, the carbonyl bond (C=O) at 1730 cm^−1^, the stretching vibrations of the phosphate group (P=O) at 1270 cm^−1^, and the P-O-Me (Me: Zn or Mg) group at 963 and 1075 cm^−1^ [14,24,25].

The surface analysis of the alloys in Figure 5a,b shows that the entire surface was affected uniformly by the alkali treatment and grafting. At higher magnification, it is noticed that the anchoring surface morphology of the as-grafted Mg alloy is mainly composed of a nanoporous hierarchical structure, while that of the Zn alloy exhibits highly rough islands. The cross-sectional morphology and corresponding elemental mapping distribution of the grafted alloys proves the existence of a composite interlayer of the initiator mainly consisting of Mg or Zn together with carbon and oxygen elements (Figure 5e). The thickness of the interlayer created is 138 ± 63 nm. After polymerization, FTIR detected peaks of asymmetric bonds of the methylene group (CH_2_) and methyl group (CH_3_) was detected at 2900 cm^−1^, carbonyl bond (C=O) at 1730 cm^−1^, and stretching vibrations of alpha methyl group (α-CH_3_) at the range from 1382 to 747 cm^−1^ (Figure 4) verifying the successful formation and growth of the PMMA film [14]. The surface morphology of the polymer-coated alloys (Figure 5d) indicates that the polymer covers the entire surface, with nanovoids detected at discrete positions.

The cross-sectional morphology and corresponding elemental mapping distribution of the polymer-coated alloys shown in Figure 5f, exhibits the formation of a densely adhered PMMA layer of an interlayer between the polymer and the substrate, exhibiting no breaking points. The thickness of the PMMA layer built is 1.7 ± 0.5 µm (Figure 5e,f), which is remarkably higher than that attained by spin coating on ZK60 magnesium alloy (470 nm) [15]. The area analysis of the coated surface that was performed through EDX shows the presence of the organic elements C and O, indicating the presence of the polymer (Figure 5c–h).

### 3.3. Electrochemical Behaviour

Representative potentiodynamic polarisation curves are shown in Figure 6a. These curves were obtained after the open circuit potentials were stabilised in SBF for 10 min and were extrapolated by the Tafel method to extract the electrochemical parameters in Table 1. The corrosion potentials (E_corr_) were ranked as follows: pure Zn (−0.89 V) < coated Zn-1.89Mg (−1.00 V) < Zn-1.89Mg (−1.22 V). The lowest E_corr_ of the uncoated Zn-1.89Mg implies the most degradation affinity. It is noticed that these values of E_corr_ of the uncoated samples are slightly lower than the initial OCP (−1.03 V for pure Zn and −1.58 V for the Zn-1.89Mg); one can refer to this as the dissolution of the naturally developed layer of the corrosion products on the samples. However, for the coated sample, the values of E_corr_ and OCP were the same to the nearest two decimal places, which means a uniform electrochemical property at the sample/electrolyte interface [26].

The corrosion current density (i_corr_), which directly influences the reaction activity and consequently the degradation rate, is ranked as: pure Zn (0.4 µA/cm^2^) < coated Zn-1.89Mg (1.1 µA/cm^2^) < Zn-1.89Mg (1.2 µA/cm^2^). For pure Zn, a repetitive cathodic slight change is detected, indicating a successive oxygen and water reduction. For the coated Zn-1.89Mg, a sharper cathodic plateau is observed, indicating a transition from an activation region near E_corr_, followed by a diffusion area, and finally a new activation region. However, for the uncoated Zn-1.89Mg, passivation is observed in the anodic branch along with that observed in the cathodic branch, which indicates a partial reduction of oxides and the formation and breakdown of the corrosion protective layer, which is mainly composed of oxides and hydroxides. The pitting is more likely to be formed when the anodic changes happen. This is attributed to the Cl- ions attacking the defects generated in the corrosion protective layer at the breakdown potentials [27]. A more detailed study of the corrosion layer is discussed later. For the three samples, passivation is observed in the range between 0.4 and 5.2 µA/cm^2^. Generally, the cathodic part of the Tafel curve is related to the hydrogen evolution part of the reaction, if it exists, whereas the anodic one is related to the metal dissolution [28].

For a comprehensive understanding of the degradation behaviour, electrochemical impedance spectroscopy (EIS) was carried out. The fitting of the Nyquist plots in Figure 6b was chosen according to circuits models represented in Figure 6c. R_s_ and R_ct_ represent the solution resistance and charge transfer resistance, respectively, whereas C_ct_ and W represent the double layer capacity and the Warburg element (for diffusion processes), respectively [29]. It must be noted that the circuit used for the fitting of the uncoated samples is different from that used for the coated ones, as the presence of the PMMA layer was considered when calculating the resistance and the double layer capacity, R_1_ and C_1._ The compound interlayer of the grafted samples was considered using the resistance and the double layer capacity, R_2_ and C_2_.

Generally, as the R_ct_ value becomes higher, the degradation rate becomes lower. The R_ct_ values in Table 1 attained by the semicircle diameter of the Nyquist plot can be arranged in a descending order as follows: 840 ± 35 Ω·cm^2^ for pure Zn, 692 ± 88 Ω·cm^2^ for coated Zn-1.89Mg, and 546 ± 101 Ω·cm^2^ for the uncoated Zn-1.89Mg. The corrosion resistance obtained from the EIS Nyquist spectra appears to be consistent with the I_corr_ and corrosion rates obtained from the potentiodynamic curves.

### 3.4. Immersion Test

Figure 7 displays the degradation rates of pure Zn, Zn-1.89Mg, and coated Zn-1.89Mg, according to the mass loss calculation after 6, 24, 120, 240, 480, and 720 h of immersion in SBF. At the all-time stage, the alloy degraded at rates much greater than the pure metal, while the coating lowered these rates to levels even lower than the pure metal. The corrosion rates of pure Zn vary between 0.09 mm/y (at 6 h) and 0.42 mm/y (at 720 h), while those of Zn-1.89Mg range between 0.09 mm/y (at 6 h) and 2.59 mm/y (at 720 h), and those of coated Zn-1.89Mg range between 0 mm/y (at 6 h) and 0.11 mm/y (at 240 h). It should be noted that extended immersion times, in contrast to the pure metal and the coated alloy, resulted in a significant and dramatic increase in corrosion rates for the uncoated alloy.

The corroded surfaces, after immersion for 120 h and 720 h in SBF, were investigated at the macro and micro levels to deeply investigate the morphology and composition of the formed corrosion layer.

The pure Zn, after 120 h of immersion, has two phases covering its surface area: a light phase (Figure 8a, point 1) and a dark one (Figure 8a, point 2). The light phase covers most of the surface, and the EDX analysis Appendix A) shows that it is composed of Zn, Cl, and O, and it is expected to be ZnO or Zn(OH)_2_ and Zn_5_(OH)_8_Cl_2_. Due to its thermal instability, the Zn(OH)_2_ phase is not likely to be formed [27,30]. The dark phase has the shape of large spots composed of small spheres with small cracks among the spheres. The dark phase is composed of only Zn and O.

The Zn-1.89Mg alloy’s surface corrosion resembles, to a great extent, that of pure Zn; however, the dark spots cover a larger area even though they are smaller in size (Figure 8b). The light phase on Zn-1.89Mg is composed of Ca and P beside the Zn, Cl, and O (Figure 8b, point 3). This is referring to the ability of Zn-1.89Mg, unlike the pure Zn, to form Ca_3_(PO_4_)_2_ and Zn_3_(PO_4_)_2_·4H_2_O compounds after immersion for 120 h in SBF. Beside the Zn and O, a small percentage of Mg was detected in the dark phase of Zn-1.89Mg (Figure 8b, point 4). For the coated Zn-1.89Mg, some compounds seemed to be deposited on the surface of the coating (Figure 8c). The deposited compounds mainly comprise Ca and P with an atomic ratio of 1.43 and a small % of Cl (Figure 8c, point 6). The two main components of the coating, C and O, exist on the whole surface in different ratios (Figure 8c, point 5). It is worth noting that no base metal constituent, Zn or Mg, was detected on the surface of the coated Zn-1.89Mg after immersion for 120 h.

Most of the surface of pure Zn after immersion in SBF for 720 h (Figure 9a) was covered by a bright phase that, according to the EDX Appendix A), is composed of Zn, O, P, and Ca, while relatively lower amounts of C and Cl were detected. This may indicate that after 720 h of immersion, the phases Ca_3_(PO_4_)_2_ and Zn_3_(PO_4_)_2_·4H_2_O could be formed on the surface of pure Zn [27,31].

The weight percent of Cl is very low compared to that on the surface of pure Zn after immersion for 240 h; this matches the results of Huang [31], who reported the depletion of Zn_5_(OH)_8_Cl_2_ at 21 days after being present for 14 days of immersion in SBF. On the surface of the corroded Zn-1.89Mg, many cracks and pores were detected between the different phases (Figure 9b). A dominant bright grey phase composed mainly of Ca, P, and O (91.62% by wt.) with a 1.23 atomic ratio of Ca to P was detected, which assures that this phase is calcium phosphate or calcium hydroxyphosphate. The macroscopic structure of the coated Zn-1.89Mg after immersion for 720 h shows a swelling on a large area of the surface (Figure 9c–e). This can be explained by previous work, where approximately 2% w/w water was found to be absorbed by PMMA [32]. With insight into the swollen area, different morphologies were identified. Some areas are composed of large, extended cracks and pores, while others consist of small spheres. The cracks are assumed to be the result of the swelling followed by dehydration as well as the cause of the leakage of the base materials, Zn and Mg, above the surface of the coating, as detected by the EDX. Notably, after 720 h of immersion, a significant amount of Ca and P was detected on the surface of the PMMA coating’s surface.

The cross-sectional investigation of the corroded samples after immersion for 720 h was carried out by SEM/EDX to study the thickness, morphology, and the attachment behaviour of the corrosion film to the base metal. It is clear from the cross-sectional presentation of the corrosion layer that the degradation of pure Zn resulted in a uniformly protective dense layer of corrosion products with no evidence for pores or cracks and of 27.5 ± 3.2 µm thickness (Figure 10a). For the uncoated Zn-1.89Mg, (Figure 10b,c), the corrosion products were sparse and formed a layer of thicknesses varying from a very few microns to 41.4 ± 3.1 µm, while the eutectic phase seems to be totally bare of any corrosion deposits. The soft α-Zn solid solution of the alloy suffers from deep pitting corrosion.

Furthermore, the two-phase laminated αZn + Mg_2_Zn_11_ eutectic structure had harsh micro galvanic corrosion with the solid solution along the whole depth of the phase. Other, more complex microgalvanic corrosion seems to be occurring between the different phases of the eutectic structure. Areas of high oxygen content on EDX (48.0 wt.% and 77.5 at.%) indicate a considered O and H_2_O reduction Appendix A). The cross-sectional representation of the coated Zn-1.89Mg in Figure 10d,e clearly exhibits the swelling which occurred in the coating and the subsequent crevice corrosion resulting from the passage of the corrosive medium to the substrate after crack formation. The crevice corrosion causes tiny pitting to grow in the base metal. No evidence of a full peeling-off of the coat was seen; instead, corrosion products formed between the coating and the substrate, which in some instances caused a dislocation of the Mg_2_Zn_11_ intermetallic precipitates. The topography of the pure Zn, Zn-1.89Mg, and coated Zn-1.89Mg surfaces after removing the corrosion products is illustrated in the Appendix A, including Appendix A.

## 4. Discussion

According to the microstructure, the ability of recrystallization in the Mg-1.89Mg was higher if compared to pure Zn. The Mg-involved phases in the alloy serve as grain nucleation sites and inhibit undesirable grain growth [26,33,34,35,36]. Furthermore, Jarzębska [22] supposes that the eutectic phase in the hypoeutectic Zn-Mg alloys results in a continuous dynamic recrystallization as a result of the construction of new grain boundaries achieved by the progressive misorientation of the adjacent sub grains. In the current study, the grain size refinement attained from recrystallization was not the main factor that affected corrosion. The corrosion was mainly governed by the composition, distribution, and morphology of the phases of the metal. This could be verified by comparing the corrosion mechanisms of pure Zn and Z-1.89Mg alloy (Figure 11).

For pure Zn, the corrosion process can be classified into stages governed by the formation of Ca/P corrosion products [37]. In the present study, for 240 h, there was no evidence for the existence of Ca or P on the surface of pure Zn. This may mean that either the immersion time was not enough for Ca/P phases to form, or that the formed Ca/P phase was not stable enough on the surface. However, the trivial change in pH suggests the first opinion. At the stages of corrosion before the Ca/P formation, the dominant reactions happening on the surface of Zn are [38]:(2)$$\mathrm{Anodic}\;\mathrm{reaction}:\;\mathrm{Zn}\to {\mathrm{Zn}}^{2+}+2e^{-}$$
(3)$$\mathrm{Cathodic}\;\mathrm{reaction}:\;2H_{2}O+O_{2}+4e^{-}\to 4{\mathrm{OH}}^{-}\;$$
(4)$$\mathrm{Overall}\;\mathrm{reaction}:\;\mathrm{Zn}+H_{2}O+O_{2}\to \mathrm{Zn}{\left(\mathrm{OH}\right)}_{2}\;$$

Due to the thermal instability of the Zn(OH)_2_ phase and the dehydration of the surface, the following reaction mostly takes place:(5)$$\mathrm{Zn}{\left(\mathrm{OH}\right)}_{2}\to \mathrm{ZnO}+H_{2}O\;$$

The aggressive chloride ions work on breaking the layer, and new compounds are formed according to the reactions:(6)$$\mathrm{Zn}{\left(\mathrm{OH}\right)}_{2}+2{\mathrm{Cl}}^{-}\to {\mathrm{ZnCl}}_{2}+2{\mathrm{OH}}^{-}$$
(7)$$5{\mathrm{Zn}}^{2+}+2{\mathrm{Cl}}^{-}+9H_{2}O\to {\mathrm{Zn}}_{5}{\left(\mathrm{OH}\right)}_{8}{\mathrm{Cl}}_{2}\cdot H_{2}O+8H^{+}$$

After that, the supersaturation of calcium and phosphates ions in the corrosive medium resulted in the following reaction:(8)$${\mathrm{Ca}}^{2+}+{\mathrm{HPO}}_{4}^{2-}+{\mathrm{xH}}_{2}O\to {\mathrm{CaPO}}_{4}\cdot {\mathrm{xH}}_{2}O$$

The calcium ions also attack the calcium phosphate compounds according to the following equation:(9)$$5{\mathrm{Ca}}^{2+}+3{\mathrm{HPO}}_{4}^{2-}+{\mathrm{Cl}}^{-}\to {\mathrm{Ca}}_{5}\left({\mathrm{PO}}_{4}\right)\mathrm{Cl}+3H^{+}$$

Reactions (8) and (9) could take place earlier on the surface of the Zn-1.89Mg alloy than pure Zn. This was proved by the 12.26 wt.% components of Ca and P found in the light phase on the surface of the Zn-1.89Mg alloy after immersion for 240 h (Figure 9b).

The homogenous Ca/P compounds can nucleate on the surface of the Zn and ZnO/Z(OH)_2_; however, the sphere-shaped Ca/P compounds make their layer inhomogeneous and allow zinc ions to escape from the base metal and to form another ZnO/Z(OH)_2_ layer on top of it. Then, reaction (7) can take place for a second time to form an additional protective layer of corrosion products [35]. However, it is reported to be unstable in the body due to the relatively high pH value and low calcium ions concentration [36]. After a sufficient time of immersion, the phases of zinc phosphates and zinc carbonates could be detected through the following reactions:(10)$$3{\mathrm{Zn}}^{2+}+2{\mathrm{HPO}}_{4}^{2-}+2{\mathrm{OH}}^{-}\to {\mathrm{Zn}}_{3}{\left({\mathrm{PO}}_{4}\right)}_{2}{\left(H_{2}O\right)}_{2}\;$$
(11)$${\mathrm{Zn}}_{5}{\left(\mathrm{OH}\right)}_{8}{\mathrm{Cl}}_{2}H_{2}O+2{\mathrm{HCO}}_{3}^{-}\to {\mathrm{Zn}}_{5}{\left({\mathrm{CO}}_{3}\right)}_{2}{\left(\mathrm{OH}\right)}_{6}+2H_{2}O+2{\mathrm{Cl}}^{-}$$

It is very clear from the morphology of the corrosion products that the degradation of pure Zn is homogenous through its entire grain (Figure 10a). There is no evidence of a selected attack at the grain boundaries. It is believed that the alloyed Zn has the same corrosion kinetics as the pure Zn; nevertheless, the existence of a second phase plays a role in promoting corrosion. The size, distribution, and volume fraction of the secondary phases define their role to work as cathodic sites throughout the biodegradation process. In this study, the eutectic sites have a clear role in creating a microgalvanic corrosion with the solid solution. Additionally, as noticed from the analysis of the alloy corrosion products, Mg has a considerable share (point 3 in Figure 9b). It is thought that the presence of Mg in the corrosion layer is unstable and less protective against further corrosion due to the interactions between the medium and Mg and the secondary interactions that occur between the components of the eutectic structure.

As a result, the corrosion layer (Figure 10b,c) was so loose that it completely split up from the eutectic sites and had a high affinity for O (point 2 in Figure 10c). Removing the corrosion products left a rough surface of 8 µm and 10 µm above and below the arithmetic mean, respectively.

Results obtained from the SEM indicate that the PMMA coating provides complete protection to the Zn-1.89Mg substrate up to 120 h of immersion in SBF and possibly to subsequent immersion periods. However, inspection of the coat after 720 h of immersion in SBF showed a swelling that appeared at the top layer of the coating as a result of polymer hydration with pores. These pores present a route for the permeability of corrosive media to the substrate surface as shown in Figure 11. The penetrating SBF resulted in crevice corrosion, and corrosion products composed of Zn and Mg are formed resulting in a blistering of the coat. Unlike the coated surface at 120 h, the EDX detected some elements from the metallic substrates in the corrosion compounds of the coated surface after 720 h of immersion. When the coating starts to absorb SBF, ester bonds are hydrolysed and acid monomers are released, resulting in the destruction of the PMMA coating. As a thin polymeric coating was investigated for this work, the release of the coating’s acidic products had a negligible effect on the medium’s pH; hence, there is no evidence of a major acceleration of zinc degradation after the coating’s disintegration.

Due to the paucity of research on Zn coatings for biodegradable applications, it is difficult to assess the efficiency of different coatings in reducing the degradation rate. However, it is allowable to compare the work with previous attempts at achieving low corrosion rates that suit biodegradable applications. Mostaed et al. [7] presented the corrosion rates of Zn alloys with relatively high contents of Mg (1% and 3% wt.) after immersion in SBF for 336 h. The results show a lower corrosion rate for the two alloys prepared by Mostaed (Zn-1Mg and Zn-3Mg) compared to the Zn-1.89Mg included in the current study. However, the coating effect was noticed to decrease the corrosion to a rate lower than the lowest corrosion rate attained by the Mostaed study, approximately 0.14 mm/y for the Zn-0.15Mg alloy [7].

Another study conducted by Yang et al. [4] introduced the corrosion rates of Zn alloyed with relatively low percentages (<1%) of Mg, Ca, Sr, Li, Mn, Fe, and Cu after immersion in SBF for 720 h. The corrosion rates of the alloys in the Yang study were all significantly lower than the alloy presented in the current study, whether coated or uncoated.

In brief, the effect of coating on suppressing the corrosion of Zn-1.89Mg is obvious. However, the coat application process should be more manipulated to attain a more uniform polymer layer. Additionally, it is believed that trying the coating on a Zn/Mg alloy with a lower content of Mg will be more effective from the point of view of corrosion prevention. This is because the magnesium content in the α-solid solution is more soluble, the eutectic phase is less enclosed, and the coating on the substrate has a more homogeneous thickness.

## 5. Conclusions

The pure extruded Zn has a satisfactory corrosion rate to be used as a biodegradable material for orthopaedic applications; however, its insufficient mechanical properties must be urgently improved. According to the literature, the alloying of Zn with Mg at low levels improves mechanical performance but significantly worsens corrosion behaviour. Indeed, pure Zn has a relatively low corrosion rate with a perfectly homogeneous corrosion behaviour. Alloying Zn with a low proportion (1.89 wt.%) of Mg results in severe localized, galvanic, and pitting corrosion, as well as a form of intergranular corrosion.

Using the “grafting from” method, a thin layer of PMMA coating measuring 1.7 ± 0.4 µm was successfully formed on the alloy substrate, which ensures the formation of strong covalent bonds between them. With this new type of coating, the corrosion rate of Zn-1.89Mg was greatly reduced, and according to the potentiodynamic test, the corrosion rate was decreased to a large extent (from 0.37 ± 0.14 mm/year without coating to 0.22 ± 0.01 mm/year with PMMA coating).

According to the immersion test, the corrosion rate decreased from 2.59 ± 1.14 mm/y to 0.09 ± 0.02 mm/y after 720 h of immersion, which is even lower than the corrosion rate of pure Zn for the same period of immersion. A full substrate protection has been established for 240 h immersion, and the protection time may be longer. After 720 h of immersion, the presence of Zn and Mg on the surface of the coated alloy varies from 0% to 24% by weight.

The surface of the alloy coated with PMMA after immersion for 240 h remains compact, but a deposit of Ca/P compound is suspected to have formed on the surface. After 720 h, swelling of the coating with cracks and pores was detected at different areas of the surface, allowing biodegradation to occur.

## Figures and Tables

**Figure 1 materials-16-00707-f001:**
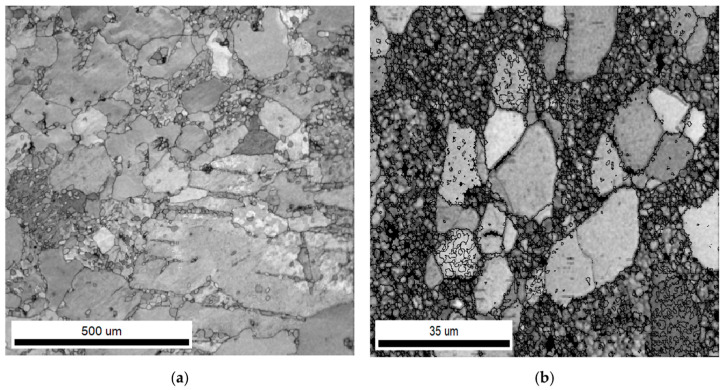
Optical microscopy of the cross-section of as-extruded: (**a**) pure Zn; (**b**) Zn-1.89Mg. Note the different scale bars.

**Figure 2 materials-16-00707-f002:**
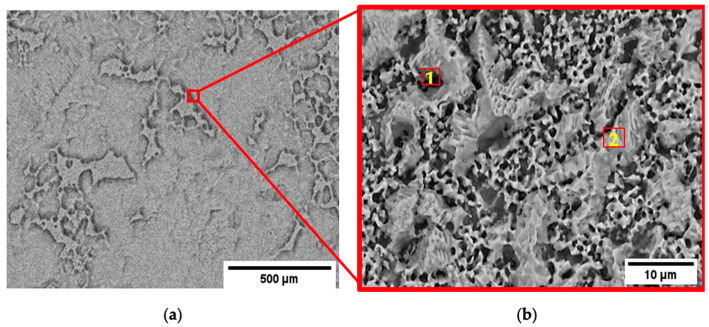
SEM/EDX analysis of the as extruded Zn-1.89M: (**a**) microstructure; (**b**) magnified window. Numbers give indications for the different phases present, EDX analysis in Appendix A.

**Figure 3 materials-16-00707-f003:**
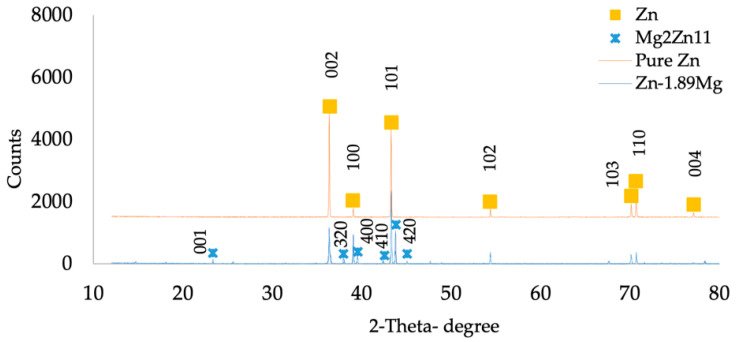
XRD analysis of pure Zn and Zn-1.89Mg.

**Figure 4 materials-16-00707-f004:**
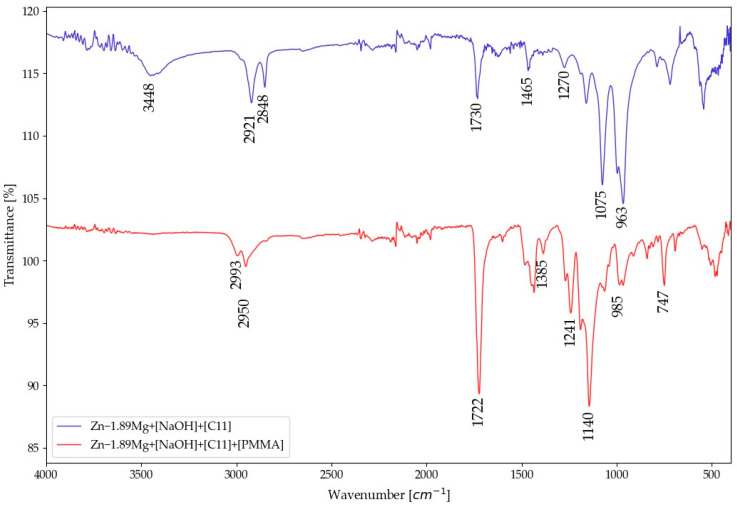
FTIR spectra of initiator-functionalized and -polymerised PMMA on the surface of the Zn-1.89Mg.

**Figure 5 materials-16-00707-f005:**
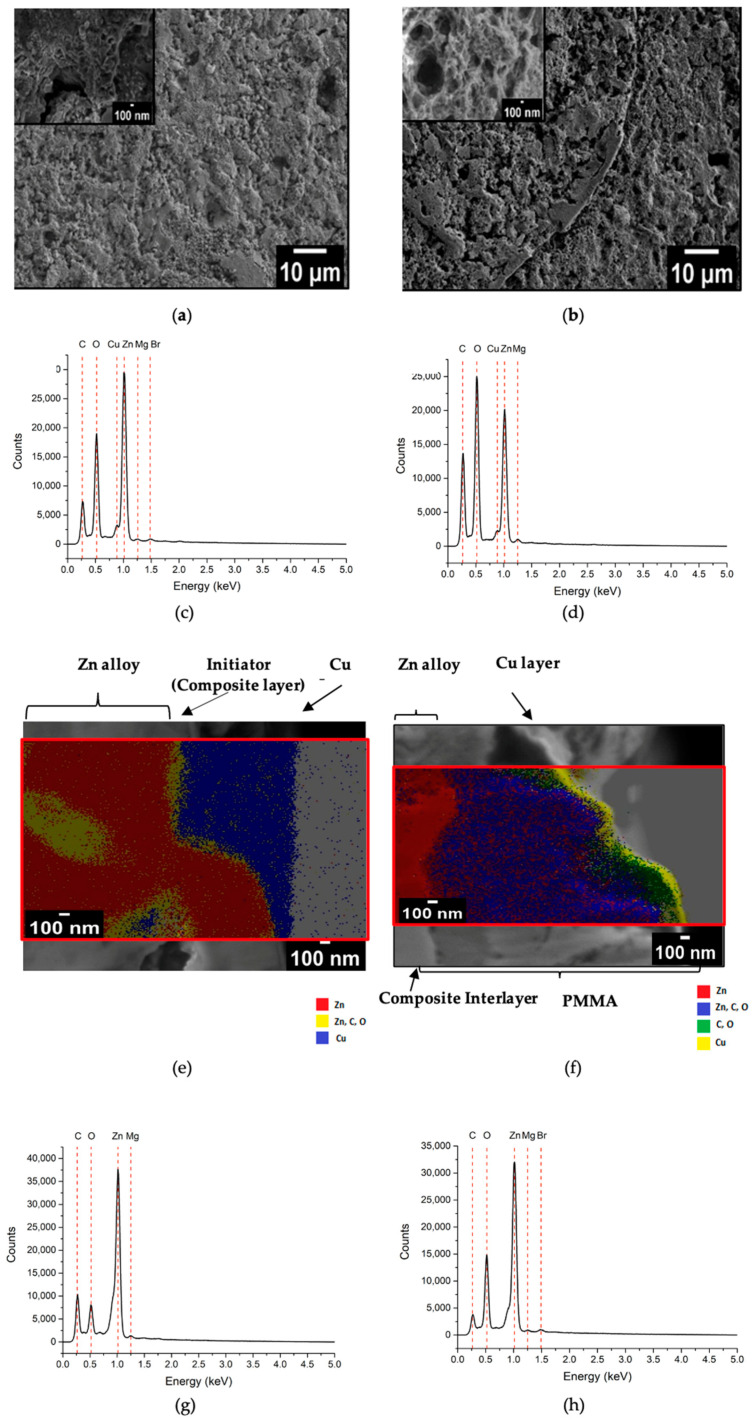
SEM/EDS/elemental mapping of as-coated Zn-1.89Mg: (**a**,**c**) surface after grafting; (**b**,**d**) surface after polymerization; (**e**,**g**) cross-section after grafting; (**f**,**h**) cross-section after polymerization.

**Figure 6 materials-16-00707-f006:**
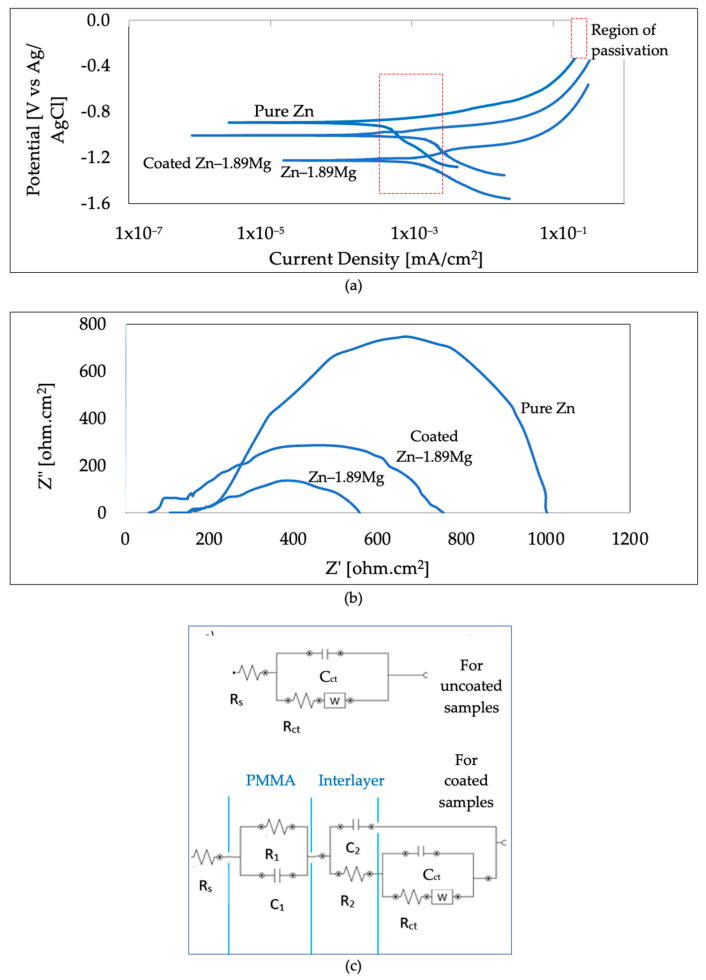
The electrochemical corrosion measurements of studied samples: (**a**) polarization curves; (**b**) Nyquist plots; (**c**) schematic representation of the used equivalent circuit.

**Figure 7 materials-16-00707-f007:**
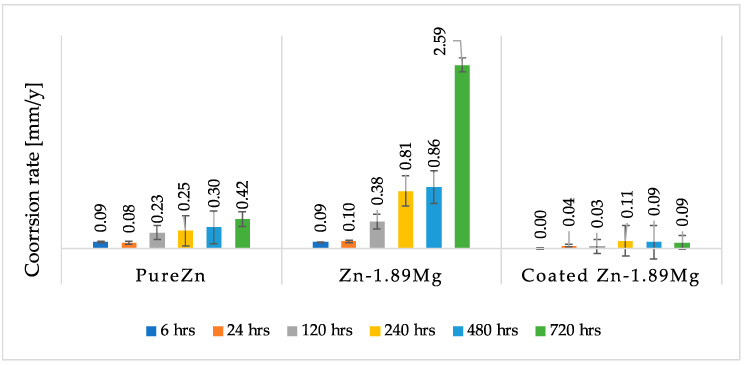
Corrosion rate variations of studied samples immersed in SBF for 6, 24, 120, 240, 480, 720 h.

**Figure 8 materials-16-00707-f008:**
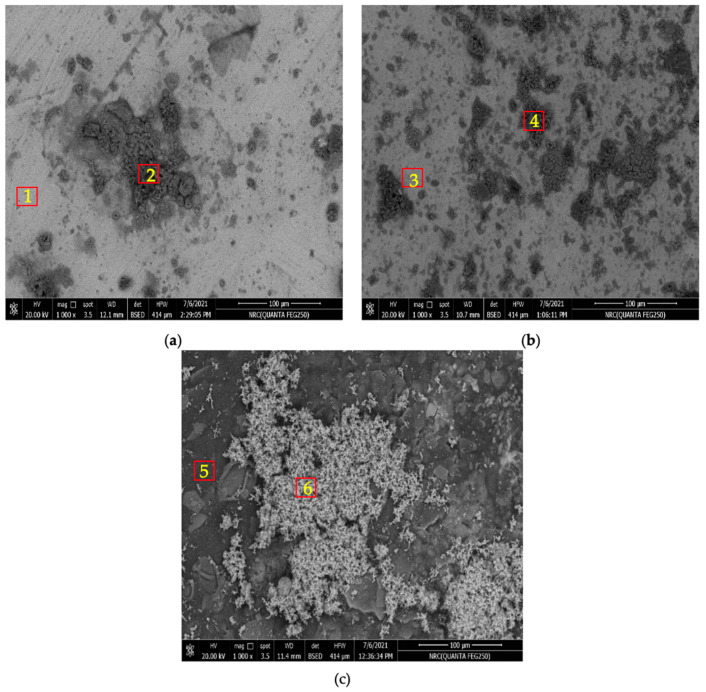
Corroded surface morphology of: (**a**) pure Zn; (**b**) Zn-1.89Mg; (**c**) coated Zn-1.89Mg after 120 h of immersion in SBF. Numbers give indications for the different phases present, EDX analysis in Appendix A.

**Figure 9 materials-16-00707-f009:**
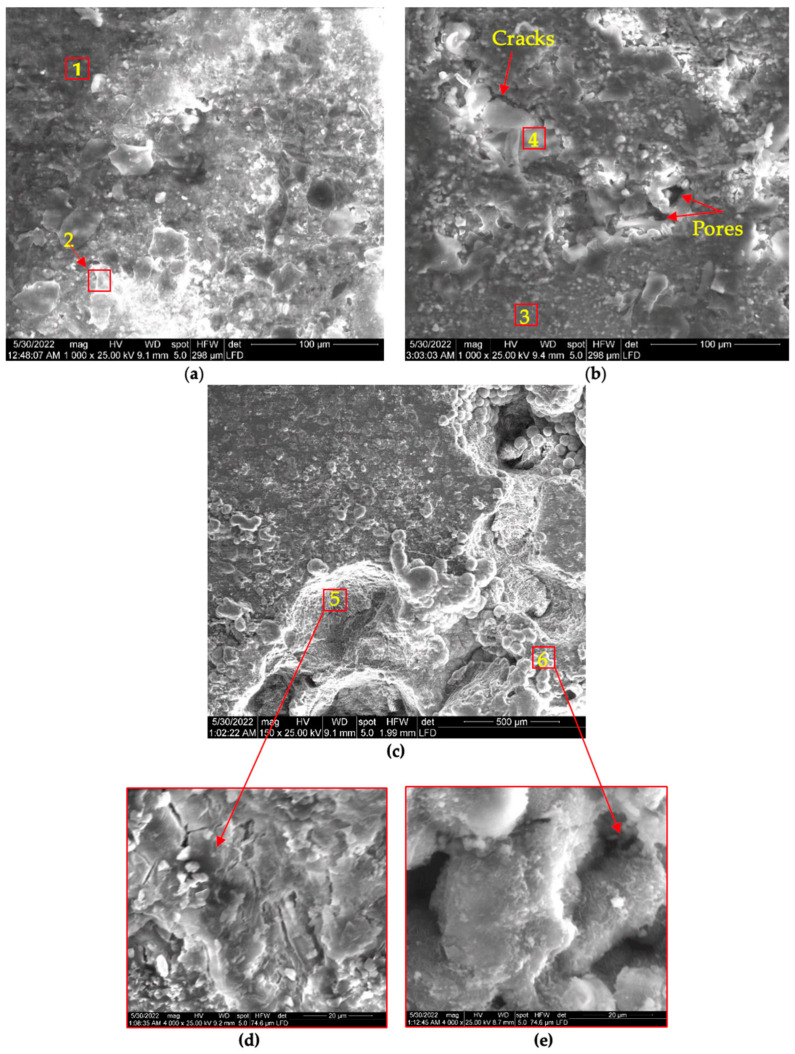
Corroded surface morphology (after 720 h of immersion in SBF) of: (**a**) pure Zn; (**b**) Zn-1.89Mg; (**c**) coated Zn-1.89Mg; and (**d**,**e**) are higher magnification for rectangles in (**c**)—note the different scale bars for better representation of the surface features. Numbers give indications for the different phases present, EDX analysis in Appendix A.

**Figure 10 materials-16-00707-f010:**
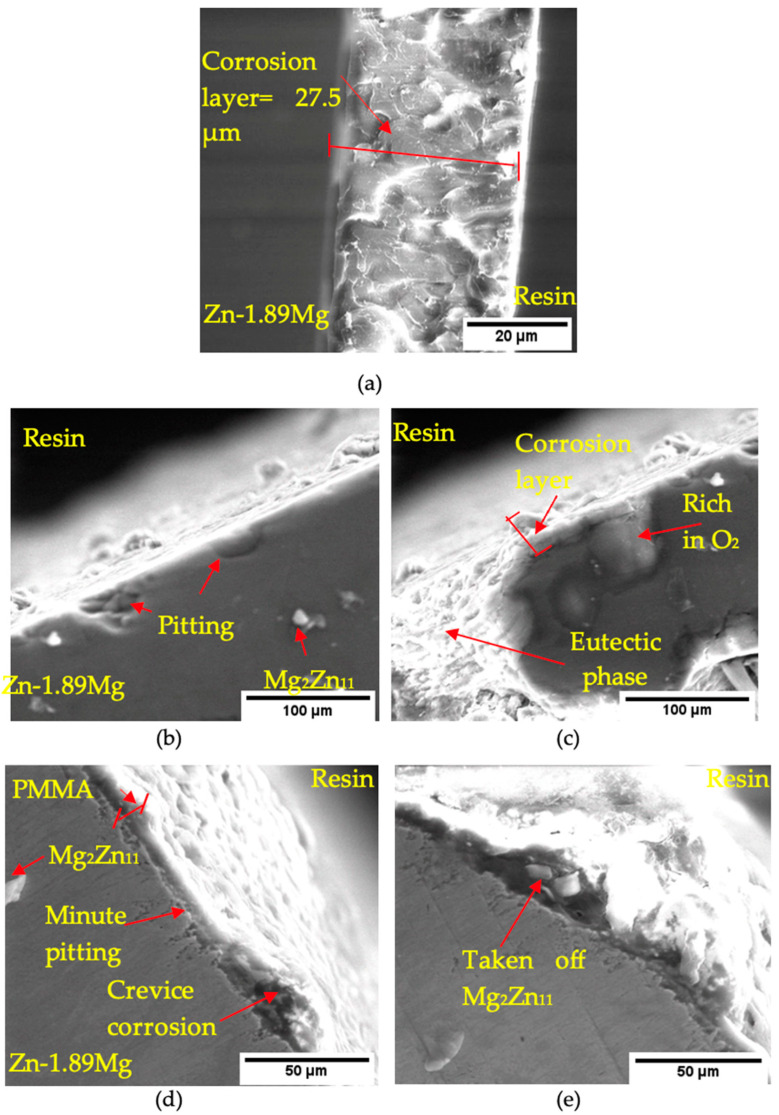
Cross-section (after 720 h of immersion in SBF) of: (**a**) pure Zn; (**b**,**c**) Zn-1.89Mg; (**d**,**e**) coated Zn-1.89M. Note the different scale bars for better representation of the features.

**Figure 11 materials-16-00707-f011:**
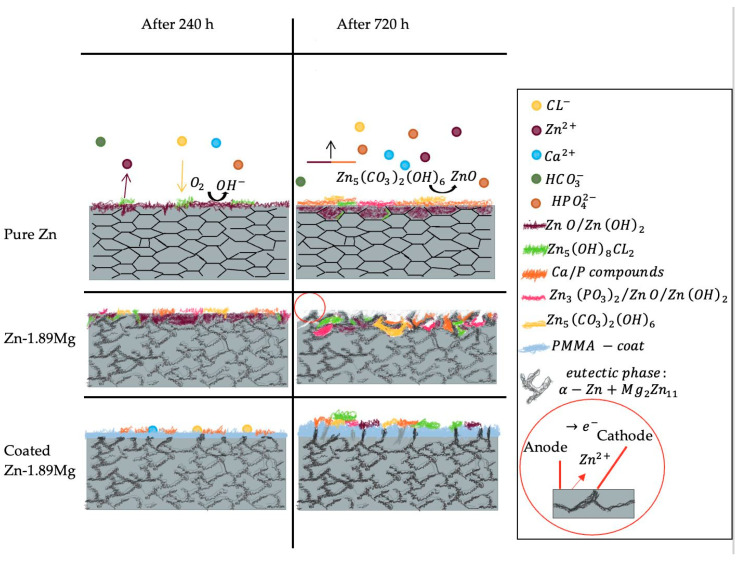
Schematic diagrams of the attack evaluation of pure Zn, Zn-1.89Mg, and PMMA-coated Zn-1.89Mg during immersion in SBF.

**Table 1 materials-16-00707-t001:** The potentiodynamic parameters extracted from the Tafel and EIS circuits fit.

Alloy	Tafel Fitting					EIS Circuit Fitting	
	Β_a_	Β_c_	E_corr_	I_corr_	Corr. Rate	R_s_	R_ct_
	(V/Dec)	(V/Dec)	(V)	(μA/cm^2^)	(mm·y^−1^)	Ω·cm^2^	Ω·cm^2^
Pure Zn	0.46 ± 0.02	0.08 ± 0.01	−0.89 ± 0.00	0.4 ± 0.1	0.12 ± 0.03	160 ± 24	840 ± 35
Zn-1.89Mg	0.34 ± 0.02	0.12 ± 0.04	−1.22 ± 0.01	1.2 ± 0.3	0.37 ± 0.14	195 ± 22	546 ± 15
PMMA Coated	0.32 ± 0.03	0.10 ± 0.01	−1.00 ± 0.01	1.1 ± 0.1	0.22 ± 0.01	55 ± 10	692 ± 88

## Data Availability

Not applicable.

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
