# Peer review of "Effect of Mg Addition and PMMA Coating on the Biodegradation Behaviour of Extruded Zn Material"

_materials, 2023, doi:10.3390/ma16020707_

Round 1

Reviewer 1 Report

The introduction fails the present the state of knowledge in the literature concerning the subject and study area of this manuscript and fails to identify a gap that this manuscript proposes to contribute. Therefore, please survey the following steps: (a) deliver background information and set the context, (b) present the specific topic of your research and clarify why it is important, (c) indicate past attempts to resolve the research problem or to response the research question and d) complete the Introduction by mentioning the specific objectives of your research. Please apply.

Materials and Methods. Add the complete instrumental details used in this current study, such as Instruments manufacturer city and country, etc., Please uniformize. More details regarding the chemical composition by ICP-AES should be introduced. A single paragraph for each characterization technique is sufficient.

Discussion. The discussion presented is poor, in terms of discussing its results and comparing them with the bibliography. I suggest reviewing this part more carefully and discuss further. Example: this section should be better presented in order to highlight the most significant and unexpected results, identify correlations, patterns and relationships among the data, speculations, limitations of work and deductive arguments.

Results. XRD – Please indicate the XRD file of the identified phases and Miller indices. Please elaborate. Please improve the quality of figure. Example: Figure 3. Please use black Font Color and indicate with a and the samples code. Please merge Figure 4 and b should be merged using an appropriate software. Please apply. For SEM images, the scale bar is not visible. Figures 3 and 7. Please use black Font color for Axis title, etc.

Conclusion part should be rewritten’ shorten to show what is the significance of your work for the study and to go beyond the results sections for forming the conclusions.

Improvements in research hypothesis, data processing and research presentation, would help to raise the manuscript quality for publication. Nevertheless, the efforts of performing all the experiments have been significant and I hope that in the near future all the issues will be solved.

Finally, I consider that the work is not suitable for publication in this form and requires large additions. If the manuscript will not be considerable improved, I will not recommend its publication.

Author Response

To the Referee,

Thank you for your patience and skill in identifying missing points.

Your contribution has allowed us to significantly enhance our paper, making it more engaging and helpful to the scientific community.

Please pardon our use of the English language, since none of us are native speakers.

The article was edited by a native English speaker.

You will find the answers to your queries listed below.

Kind regards

Adele Carradò

___________________________________________________________________

Referee 1

The introduction fails the present the state of knowledge in the literature concerning the subject and study area of this manuscript and fails to identify a gap that this manuscript proposes to contribute. Therefore, please survey the following steps: (a) deliver background information and set the context, (b) present the specific topic of your research and clarify why it is important, (c) indicate past attempts to resolve the research problem or to response the research question and d) complete the Introduction by mentioning the specific objectives of your research. Please apply.

Answer: The introduction was modified according to the referee`s guidance.

Materials and Methods. Add the complete instrumental details used in this current study, such as Instruments manufacturer city and country, etc., Please uniformize. More details regarding the chemical composition by ICP-AES should be introduced. A single paragraph for each characterization technique is sufficient.

Answer: The instrumental details in the materials and method section was modified according to the referee`s guidance.

Discussion. The discussion presented is poor, in terms of discussing its results and comparing them with the bibliography. I suggest reviewing this part more carefully and discuss further. Example: this section should be better presented in order to highlight the most significant and unexpected results, identify correlations, patterns and relationships among the data, speculations, limitations of work and deductive arguments.

Answer: The discussion was modified according to the referee`s guidance.

Results. XRD – Please indicate the XRD file of the identified phases and Miller indices. Please elaborate. Please improve the quality of figure. Example: Figure 3. Please use black Font Color and indicate with a and the samples code. Please merge Figure 4 and b should be merged using an appropriate software. Please apply. For SEM images, the scale bar is not visible. Figures 3 and 7. Please use black Font color for Axis title, etc.

Answer: XRD figure was modified, and miller indices was added.

              Figures 3 was modified to increase quality and font was changed to black.

              Figure 4 a and b were merged.

              SEM figures were modified to have a visible scale bar.

              Figures 4 was modified to increase quality and font was changed to black.

Conclusion part should be rewritten’ shorten to show what is the significance of your work for the study and to go beyond the results sections for forming the conclusions.

Answer: The conclusion was modified according to the referee`s guidance.

Improvements in research hypothesis, data processing and research presentation, would help to raise the manuscript quality for publication. Nevertheless, the efforts of performing all the experiments have been significant and I hope that in the near future all the issues will be solved.

Finally, I consider that the work is not suitable for publication in this form and requires large additions. If the manuscript will not be considerable improved, I will not recommend its publication.

Author Response

To the Referee,

Thanks for your patience and skill in spotting errors. Your help has made our paper more interesting and useful to the scientific community.

We're not native English speakers, so please excuse us. English coworker edited the article.

Answers are below.

Kind regards

Adele Carradò

___________________________________________________________________

Referee 2

Introduction

  1. “Attempts have been made to develop biodegradable metals that are prospected to degrade steadily in vivo with a proper host reaction. The corrosion by-products of the biodegradable metals should be released so that cells may digest, metabolize or excrete them. Meanwhile, the biodegradable metal implant should completely disintegrate after the role of supporting it has been done.”

Please provide reference(s) to a literary source. Who made the attempts?

Answer: Reference has been assigned to the paragraph.

Results

3.1. Microstructure of pure Zn and Zn-1.89Mg

“Figure 1. Optical microscopy of the cross-section of as extruded of: (a) pure Zn; (b) Zn-1.89Mg. Note the different scale bars.”

It is logical to compare the microstructure in this figure at the same magnification. It is most optimal to present then 4 photos with two scales for pure zinc and alloy. Give explanations, please.

Answer: If a microstructure of pure Zn was introduced by an optical micrograph with the same scale of the alloy Zn-1.89Mg, only one big grain will be shown. So, a hint in the manuscript was added to draw the reader attention for the change in scale between the two microstructures.

3.2. Surface of PMMA-coated Zn-1.89Mg

In this section, the authors describe the IR spectra and relate them to specific vibrations and functional groups. However, the authors do not refer to some database or literary sources. Provide links to the database or other sources, please.

Answer: Ihe IR section was modified according to the referee`s guidance.

3.4. Immersion test

“This is referring to the ability of Zn-1.8Mg, unlike the pure Zn, to form Ca3(PO4)2 and

Zn3(PO4)2.4H2O compounds after immersion for 120 h in SBF.”

If such phases have been established in previous studies, then the literary source should be indicated. And if in this study, then it is necessary to provide XRD data. Please clarify.

Answer: Phases have been established in previous studies and for clarification references were added in the manuscript to.

“Figure 8.”

In figures 8 a, b, odd numbers correspond to light areas of the surface, and even numbers - to dark ones. Therefore, it would be logical to change the numbers 5 and 6 in figure c as well.

Answer: The numbers 5 and 6 in figure 8c were changed, and, accordingly changed in the supplementary materials: Table S2: Elemental analysis of the points 1 - 6 in Figure 8.

Round 2

Reviewer 1 Report

To my surprise authors did great job very shortly. The authors have made corrections regarding all my comments. I recommend the article for publication.